# Dysregulation of Immune Cell Subpopulations in Atypical Hemolytic Uremic Syndrome

**DOI:** 10.3390/ijms241210007

**Published:** 2023-06-11

**Authors:** I-Ru Chen, Chiu-Ching Huang, Siang-Jyun Tu, Guei-Jane Wang, Ping-Chin Lai, Ya-Ting Lee, Ju-Chen Yen, Ya-Sian Chang, Jan-Gowth Chang

**Affiliations:** 1Graduate Institute of Clinical Medical Science, College of Medicine, China Medical University, Taichung 40402, Taiwan; angiogenesisbody@gmail.com (I.-R.C.); drcchhuang@gmail.com (C.-C.H.); jennyw355@gmail.com (G.-J.W.); 2Division of Nephrology and the Kidney Institute, Department of Internal Medicine, China Medical University and Hospitals, Taichung 404332, Taiwan; georgepclai@gmail.com; 3Center for Precision Medicine, China Medical University Hospital, Taichung 404332, Taiwan; mist0205@gmail.com (S.-J.T.); t23701@mail.cmuh.org.tw (Y.-T.L.); t24399@mail.cmuh.org.tw (J.-C.Y.); t25074@mail.cmuh.org.tw (Y.-S.C.); 4Department of Medical Research, China Medical University Hospital, Taichung 404332, Taiwan; 5Epigenome Research Center, China Medical University Hospital, Taichung 404332, Taiwan

**Keywords:** atypical hemolytic uremic syndrome, complement, disease activity, single cell sequencing, therapy

## Abstract

Atypical hemolytic uremic syndrome (aHUS) is a rare, life-threatening thrombotic microangiopathy. Definitive biomarkers for disease diagnosis and activity remain elusive, making the exploration of molecular markers paramount. We conducted single-cell sequencing on peripheral blood mononuclear cells from 13 aHUS patients, 3 unaffected family members of aHUS patients, and 4 healthy controls. We identified 32 distinct subpopulations encompassing 5 B-cell types, 16 T- and natural killer (NK) cell types, 7 monocyte types, and 4 other cell types. Notably, we observed a significant increase in intermediate monocytes in unstable aHUS patients. Subclustering analysis revealed seven elevated expression genes, including *NEAT1*, *MT-ATP6*, *MT-CYB*, *VIM*, *ACTG1*, *RPL13*, and *KLRB1*, in unstable aHUS patients, and four heightened expression genes, including *RPS27*, *RPS4X*, *RPL23*, and *GZMH* genes, in stable aHUS patients. Additionally, an increase in the expression of mitochondria-related genes suggested a potential influence of cell metabolism on the clinical progression of the disease. Pseudotime trajectory analysis revealed a unique immune cell differentiation pattern, while cell—cell interaction profiling highlighted distinctive signaling pathways among patients, family members, and controls. This single-cell sequencing study is the first to confirm immune cell dysregulation in aHUS pathogenesis, offering valuable insights into molecular mechanisms and potential new diagnostic and disease activity markers.

## 1. Introduction

Atypical hemolytic uremic syndrome (aHUS) is a rare and life-threatening thrombotic microangiopathy (TMA) disease characterized by a triad of microangiopathic hemolytic anemia, thrombocytopenia, and acute kidney injury. This presentation is distinct from thrombotic thrombocytopenic purpura and other TMA diseases. The TMA of aHUS affects multiple organ systems, often leading to rapid multi-organ failure and mortality [1]. The pathogenesis of aHUS is closely related to the dysregulation of the complement system, which normally functions to protect the body against invading pathogens that can damage cells. In aHUS, alternative complement pathway gene mutations or the dysregulation of complement regulators can lead to the excessive activation of the complement system after the body’s environment has come into contact with a range of triggers, including infection, pregnancy, and specific drugs. The deposition of complement components on endothelial cells triggers a series of downstream events, including the recruitment and activation of immune cells, such as neutrophils and monocytes, which adhere to the endothelial cells and contribute to the formation of blood clots. This process is mediated by the expression of adhesion molecules on the surface of endothelial cells, which interact with receptors on the surface of immune cells [2]. Furthermore, mutations in the genes related to the lectin pathway, classic pathway, and coagulation pathway have been identified in several aHUS studies. It should be noted that patients may present with multiple genetic variations, as opposed to a single gene mutation [3]. Plasma exchange is not universally effective in all cases of aHUS, and a significant number of aHUS patients require costly anti-complement therapy as a life-saving measure. The optimal timing for the discontinuation of anti-complement therapy in aHUS patients remains undefined. However, the monitoring of hemolysis markers—including hemoglobin, platelet count, lactate dehydrogenase, haptoglobin, blood smear, and an evaluation of organ function, as well as the measurement of 50% hemolytic complement (CH50) activity in serum—may be used to guide this decision [4].

Complement plays a critical role in regulating the activation of T- and B-cells, and this is achieved through various receptors and regulators, such as CD46, CR1, CD59, and CD55. C3d acts as a molecular adjuvant for B-cell activation by reducing the activation threshold. Furthermore, the interaction of CR2 on B-cells with C3d and iC3b forms a co-receptor complex with CD19 and CD81, which facilitates antigen presentation and the production of antigen-specific IgG. C4 is important for maintaining B-cell tolerance, while the activation of complement may lead to autoimmune diseases. C3 is also essential for the generation and persistence of memory B-cells in germinal centers, where it binds to CR2 on follicular dendritic cells, leading to the presentation of antigens and the induction of effector and memory B-cells [5].

Approximately 35–50% of patients with atypical hemolytic uremic syndrome (aHUS) do not exhibit any detectable pathogenic genes [4,6]. However, these patients still respond to anticomplement therapy. Currently, there is no definitive biomarker to indicate the disease activity of aHUS. Clinicians must rely solely on the improvement of clinical symptoms of TMA in the absence of the progression of hemolysis markers. Unfortunately, in critical clinical scenarios, such indicators may only manifest after it is too late to control the progression of aHUS. Therefore, the identification of a molecular marker for disease activity is of utmost importance. Given the critical role of complement in maintaining the homeostasis of the human immune system and endothelial adhesion—and the correlation with immune cells—while the current understanding of the circulating immune cell types and states implicated in aHUS remains unknown, we hypothesize that the dysregulation of specific immune cells may contribute to the pathogenesis of aHUS.

Single-cell sequencing (scRNA-seq) is a cutting-edge tool that permits the assessment of heterogeneity among various immune subpopulations, enabling the investigation of gene expression and intracellular signaling pathways at the single-cell or cell type level to understand the progression of diseases. To explore the role of immune cell dysregulation in aHUS pathogenesis, we conducted scRNA-seq analysis on peripheral blood mononuclear cells (PBMCs) from 13 aHUS patients, 3 aHUS relative family members without a diagnosis of aHUS, and 4 healthy controls, identifying specific immune cell subpopulations and pathways implicated in aHUS. We compared immune cell dysregulation between aHUS patients, aHUS family members, and healthy controls, as well as between aHUS patients with stable versus unstable disease activity and those undergoing plasmapheresis versus those receiving both plasmapheresis and anti-complement therapy. Our study is the first to examine immune cell regulation in aHUS, deepening our understanding of immune cell pathogenesis and providing clinical insights.

## 2. Results

### 2.1. The Demography of Studied Cases

Table 1 presents the clinical features of 13 aHUS patients diagnosed with hemolytic anemia, thrombocytopenia, and acute kidney injury, excluding infectious, autoimmune, or malignant etiologies. The patients’ ages spanned from 30 to 81 years, with balanced gender distribution (7:6). All exhibited extrarenal manifestations, with nine (69.2%) stable and four (30.8%) unstable cases. Treatment involved plasma exchange alone (five cases, 38.5%) or combined with anti-complement therapy (eight cases, 61.5%).

### 2.2. The Immunological Landscape of Immune Cells from aHUS, aHUS Family, and Healthy

We conducted scRNA-seq on PBMCs from aHUS patients (N = 13), family members (N = 3), and healthy controls (N = 4) to examine immune cell heterogeneity in aHUS (Figure 1a). After preprocessing and quality control, we obtained single-cell transcriptomes of 112,191, 24,848, and 37,539 immune cells from aHUS patients, family members, and healthy controls, respectively. This enabled distinguishing among groups, disease activity, and treatments (plasma exchange only, combined anti-complement therapy).

Using SCTransform normalization and robust principal component analysis (rPCA) in Seurat, we identified 32 PBMC cell subpopulations in aHUS patients. SingleR annotation predicted B-cells, T-cells, monocytes, macrophages, dendritic cells, NK cells, megakaryocytes, granulocytes, and progenitors (Figure 1b). The analysis of five B-cell subpopulations displayed high diversity in aHUS patients and families, influenced by disease activity and treatment (Figure 2a–c). The investigation of 16 T and NK cell subpopulations also revealed significant diversity (Figure 3a), impacted by disease activity and treatment (Figure 3b,c). In seven monocyte subpopulations, aHUS patients had more intermediate monocytes than healthy controls (Figure 4a). Additionally, intermediate monocytes with higher numbers in the unstable aHUS group (Figure 4b) increased in combined treatment compared to plasma exchange alone (Figure 4c).

### 2.3. Cell Populations in PBMCs

#### 2.3.1. Comparing aHUS Patients, aHUS Family, and Healthy Controls

Utilizing Wilcoxon rank sum tests, we discovered significant increases in various immune cell populations, such as plasmablasts, intermediate monocytes, terminal effector CD4 T-cells, Th1 cells, Th1/Th17 cells, Th17 cells, effector memory CD8 T-cells, central memory CD8-T cells, and terminal effector CD8 T-cells in aHUS patients compared to the controls (*p* < 0.05, Figure 5a–i). Notably, aHUS families showed intermediate cell population levels between aHUS patients and healthy controls for plasmablasts, intermediate monocytes, terminal effector CD4 T-cells, Th1 cells, effector memory CD8 T-cells, and terminal effector CD8 T-cells.

Non-switch memory B-cells and plasmacytoid dendritic cells were more abundant in controls than in aHUS patients and families (*p* < 0.05, Figure 5j,k). Intermediate and classical monocytes were higher in patients compared to families (*p* < 0.05, Figure 5l,m), while non-classical monocytes were lower in patients compared to families (*p* < 0.05, Figure 5n).

#### 2.3.2. Comparing Stable and Unstable aHUS Patients, aHUS Family, and Healthy Controls

Intermediate monocytes were significantly enriched in the unstable aHUS group, followed by the stable group, aHUS family, and the controls (*p* < 0.05, Figure 6a). Conversely, classical monocytes were enriched in the stable group compared to the unstable group (*p* < 0.05, Figure 6b). Plasmablasts, non-Vd2 gd T-cells, and effector memory CD8 T-cells increased in the unstable group, followed by the stable group, aHUS family, and controls, with significant differences between the control and unstable groups (*p* < 0.05, Figure 6c–e).

Plasmacytoid dendritic cells were more abundant in the healthy group, followed by the aHUS family, stable group, and unstable group (*p* < 0.05, Figure 6f), with the unstable group showing significantly lower levels compared to the control group. For non-switched B-cells, the stable group had significantly lower levels compared to the control group (*p* < 0.05, Figure 6g).

#### 2.3.3. Comparing Different Treatment in aHUS Patients, aHUS Family, and Healthy Controls

In this subgroup analysis of aHUS treatment, intermediate monocyte enrichment showed an increasing trend from the plasma exchange group to the combined plasma exchange with anti-complement therapy group, aHUS family group, and healthy control group (Figure 7a). The difference was only statistically significant (*p* < 0.05) between plasma exchange and healthy control groups, with no significant difference between the two treatment groups. Plasmacytoid dendritic cell abundance exhibited an increasing trend from the healthy control group to the aHUS family group, followed by the combined therapy group and plasma exchange group, with the highest levels occurring in the controls and the lowest levels in the plasma exchange group (Figure 7b). The combined therapy group exhibited significant enrichment of follicular helper T-cells, Th1/Th17 cells, and Th17 cells compared to the plasma exchange group (Figure 7c–e, *p* < 0.05). Non-switched memory B-cells were significantly less abundant in the combined therapy group compared to the control group (Figure 7f, *p* < 0.05).

### 2.4. Cell Subclusters in PBMCs

#### 2.4.1. Comparing aHUS Patients, aHUS Family, and Healthy Controls

This study identified significant differences in immune cell subclusters among aHUS patients, aHUS family members, and healthy controls. In aHUS patients compared to healthy controls, we observed increased levels of classical monocytes (subclusters 6, 7) with higher *RPS27* and *IFI27* expression (Figure 8a,b), central memory CD8 T-cells (subcluster 3) with higher *CXCR4* expression in patients and family members, non-Vd2 gd T-cells (subcluster 4) with higher *SYNE2* expression, Th1 T cells (subcluster 3) with higher *MT-CYB* expression, and Th17 cells (subcluster 4) with higher *MT-ATP6* expression (Appendix A).

Conversely, in healthy controls compared to aHUS patients, increased levels were found for central memory CD8 T-cells (subcluster 1) with higher *EIF3E* expression, Th1 cells (subcluster 0) with higher *RPS27* expression, non-classical monocytes (subcluster 5) with higher *LYPD2* expression, terminal effector CD4 T-cells (subcluster 3) with higher *KLRD1* expression, and Th17 cells (subcluster 3) with higher *ACTG1*, *CD52*, and *LGALS1* expression (Appendix A). Gene expression levels for each cell type were illustrated using dot plots, and the findings are summarized in Table 2.

#### 2.4.2. Comparing Stable and Unstable aHUS Patients, aHUS Family, and Healthy Controls

Our study revealed significant differences in immune cell subclusters among unstable aHUS, stable aHUS, aHUS family members, and healthy controls. In unstable aHUS compared to stable aHUS, we observed increased classical monocytes (subcluster 4) with higher *NEAT1, MT-ATP6*, and *MT-CYB* expression, central memory CD8—cells (subclusters 2) with elevated *VIM* expression, non-Vd2 gd T-cells (subcluster 1) with increased *ACTG1* expression, and terminal effector CD8 T-cells (subclusters 3, 5) with elevated *RPL13* and *KLRB1* expression (Appendix A).

In contrast, subclusters that increased in stable aHUS compared to unstable aHUS include central memory CD8 T-cells (subcluster 1) with higher *RPL23* expression, non-Vd2 gd T-cells (subcluster 0) with elevated *GZMH* expression, and Th1 cells (subcluster 0) with increased *RPS27* and *RPS4X* expression (Appendix A). These findings are summarized in Table 3.

### 2.5. Trajectory Analysis for B-Cell, T-Cell, and Monocyte

#### 2.5.1. Comparing aHUS Patients, aHUS Family, and Healthy Control

Cytopath analysis showed immune cell state dynamics in B-cell, T-cell, and monocyte trajectories (Figure 9). Naïve B-cells, exhausted B-cells, and non-switched memory B-cells in the aHUS group peaked at pseudotimes 0, 9, and 12, differing from healthy controls and aHUS nuclear families. From pseudotimes 5 to 10, naïve B-cell abundance was highest in the healthy controls, followed by aHUS families, and was lowest in the aHUS group (Figure 10a).

CD4 T-cell trajectory analysis showed distinct abundance patterns for naïve CD4 T-cells, T regulatory cells, Th1 cells, and Th1/Th17 cells in healthy controls, aHUS families, and the aHUS group across pseudotimes. During 20–25, Th2, Th17, and Th1/Th17 cells were the most abundant in the healthy controls, followed by aHUS families and the aHUS group (Figure 10b). Terminal effector CD4 T-cell abundance exhibited a similar pattern in pseudotimes 35–40. At pseudotime 30, Th2, Th1, T-regulatory, and follicular T-helper cells were most abundant in the aHUS group, followed by aHUS families and the healthy controls.

In pseudotimes 0–5, naïve CD8 T-cell abundance was highest in the healthy controls, followed by aHUS families and patients (Figure 10c). This trend reversed in pseudotimes 5–10. No significant differences in plasmacytoid dendritic cells, myeloid dendritic cells, non-classical monocytes, and classical monocytes were observed among aHUS patients, families, and healthy controls in pseudotimes 0–30. However, intermediate monocyte abundance in aHUS patients significantly increased during pseudotimes 7–10, which was not observed in the families or healthy controls (Figure 10d).

#### 2.5.2. Comparing Stable and Unstable aHUS Patients, aHUS Family, and Healthy Controls

The unstable aHUS group showed a significant increase in exhausted B-cells during pseudotimes 7–13 compared to stable aHUS, aHUS families, and healthy controls, followed by a decline from pseudotimes 13–18. In this interval, switched memory B-cell abundance was lowest in the unstable group compared to the others. Non-switched memory B-cells were more abundant in the unstable group, with the largest difference at pseudotimes 10–15 (Figure 10e,f).

Th2, Th17, and Th1/Th17 cells had the lowest abundance in the unstable aHUS group at pseudotimes 18–22 but peaked at pseudotimes 28–32 for Th2, Th17, Th1/Th17, Th1, T regulatory, follicular helper T, and naïve CD4 T-cells (Figure 10f). The stable aHUS group showed a pattern more akin to the unstable group than to aHUS families and healthy controls.

### 2.6. Immune Cell Interactions in Blood Samples from aHUS, aHUS Family, and Healthy Control

To establish a comprehensive immune cell–complement pathway interaction network, we used the STRING database, integrating identified pathway interactions with aHUS-associated genes, including *CFH*, *CD46*, *CFI*, *C3*, *CFB*, *THBD*, *CFHR1-5*, *DGKE*, *VTR*, *C2*, *C3AR1*, *C8B*, *C9*, *C4BPA*, *CFD*, *MASP1-2*, *MMACHC*, *PLG*, *WT1*, *VWF*, *CR1*, *CXCL12*, *C5*, *TLR4*, *CXCR4*, *HASP*, *KNK*, *INF2*, *EXOSC3*, *TSEN2*, *CD36*, and *VTN* [2,4,5,7,8,9,10,11,12,13,14,15,16]. The enriched pathways in aHUS patients include ALCAM-CD6, IL16-CD4, APP-CD40, CD86-CTLA4, CXC, and SELPLG (Figure 11a–d and Appendix A).

ALCAM-CD6 interactions had two patterns: one resembling healthy controls and another similar to aHUS families. Plasmacytoid dendritic cells exhibited increased outgoing signaling in pattern 1, while various T-cells showed increased incoming signaling in pattern 2.

APP-D40 interactions in aHUS patients were divided into two patterns, with pattern 2 being similar to aHUS family and healthy controls. Classical monocytes, intermediate monocytes, megakaryocytes, and myeloid dendritic cells demonstrated increased outgoing signaling in pattern 1.

IL16-CD4 interactions in aHUS patients had two patterns. Outgoing signaling from myeloid dendritic cells and plasma blasts was highest in pattern 1, while intermediate monocytes, myeloid dendritic cells, non-classical monocytes, and classical monocytes exhibited increased incoming signaling.

CD86-CTLA4 pathways in aHUS patients displayed distinct patterns with heightened outgoing signaling in non-classical monocytes. In the SELPLG pathway, cases a7 and a10 showed significantly increased outgoing signaling in megakaryocytes. The CXC interactions in aHUS patients had three patterns. Patterns 1 and 2 demonstrated significantly reduced incoming signaling from non-switched memory B-cells, central memory CD8 T-cells, Vd2 gd T-cells, Th1/Th17 cells, and Th1 cells compared to pattern 3 and the healthy control. Notably, case a10 exhibited significantly increased outgoing signaling from non-classical monocytes and incoming signaling from MAIT cells, which was unobserved in other participants.

## 3. Discussion

aHUS represents a rare, life-threatening condition. The challenges in diagnosing and managing aHUS largely stem from the absence of specific diagnostic markers and the disease’s rapid progression. In a pioneering effort, our study employs single-cell sequencing to probe immune cell dysregulation in aHUS, thereby offering unique insights into the disease’s pathogenesis and its clinical implications.

In our study, we analyzed cell subpopulations and found that aHUS patients had higher levels of plasmablasts, intermediate monocytes, terminal effector CD4 T-cells, Th1 cells, effector memory CD8 T-cells, and terminal effector CD8 T-cells compared to aHUS families and healthy controls. In contrast, non-switch memory B-cells and plasmacytoid dendritic cells were most abundant in healthy controls, followed by aHUS families and patients. The unstable aHUS group showed significantly higher intermediate monocyte abundance than stable aHUS, aHUS families, and healthy controls. We suggest intermediate monocytes as potential aHUS disease activity markers. In the study conducted by Zawada AM et al., the pivotal role of monocytes is examined, where they are segmented into three distinct subsets with a particular emphasis on the potential involvement of the intermediate subset in atherosclerosis [17]. In the research presented by Wong KL et al., the authors delineate the specific phenotypes and functions of these monocyte subsets and further discuss alternative markers for their segregation [18]. Finally, Ziegler-Heitbrock L. et al. propose an officially endorsed classification for monocyte and dendritic cell subsets in their study, aiming to streamline communication and spur further research within the scientific community [19]. These monocytes also interact with endothelial cells, indicating a potential contribution to aHUS pathogenesis and correlation with endothelial cells.

In the study led by Perez RK et al., they identify a heightened expression of type 1 interferon-stimulated genes in monocytes, a decrease in naive CD4+ T-cells that was aligned with the upregulated monocyte ISG expression, and an expansion of cytotoxic GZMH+ CD8+ T-cells with limited repertoire diversity [20]. In the study led by Nehar-Belaid D et al., they notice the expansion of unique interferon-stimulated genes and/or monogenic lupus-associated gene-enriched subpopulations which could identify patients with the highest disease activity [21]. Li Y et al. demonstrates that a six-protein combination (*IFIT3*, *MX1*, *TOMM40*, *STAT1*, *STAT2*, and *OAS3*) offers valuable diagnostic utility for systemic lupus erythematosus (SLE) [22]. Zhang Y et al. further substantiate this by observing increased levels of macrophage migration inhibitory factor (MIF) in the serum of SLE patients [23]. Shifting focus to IgG4-related disease (IgG4-RD), Wu X et al. identify increased proportions of CD8 central memory T- (TCM) and TIGIT+ CD8 cytotoxic T (CTL)-cells in patients compared to the healthy controls. Their additional analysis illuminates the critical role of B-cell activation factor (BAFF) signaling pathways, showing their enrichment from myeloid cell subsets to B-cells [24].

In this study, we identified significant gene upregulation in various immune cell subclusters in aHUS patients compared to healthy controls. Classical monocyte subclusters 6 and 7 showed upregulated *RPS27* and *IFI27* genes, while central memory CD8 T-cells subcluster 3, non-Vd2 gd T-cells subcluster 4, Th1 cells subcluster 3, and Th17 cells subcluster 4 exhibited upregulated *CXCR4*, *SYNE2*, *MT-CYB*, and *MT-ATP6* genes, respectively.

We also observed distinct gene expression patterns between unstable and stable aHUS. Unstable aHUS exhibited increased expression of *NEAT1*, *MT-ATP6*, *MT-CYB*, *VIM*, *ACTG1*, *RPL13*, and *KLRB1* genes in various immune cell subclusters, while stable aHUS showed upregulated *RPS27, RPS4X, RPL23*, and *GZMH* genes. These genes may serve as potential clinical markers for aHUS disease activity. Elevated mitochondria-related gene expression suggests cell metabolism’s role in the aHUS clinical course, warranting further investigation. Notably, these gene expression patterns were not observed in other autoimmune diseases, such as systemic lupus erythematosus [20,21,22,23] or immunoglobulin G4-related disease [24], highlighting the unique immune cell profile in aHUS.

Our pseudotime trajectory analysis revealed a unique point where immune cell differentiation in aHUS patients diverged from healthy individuals. This divergence was also apparent in some aHUS family members, falling between aHUS patients and healthy controls.

In our cell–cell interaction analysis, we aimed to identify signaling pathway differences between healthy individuals and aHUS patients, ensuring observed complement and immune cell interactions were not comparable between groups. The results showed unique signaling patterns in aHUS patients, specifically in ALCAM-CD6, IL16-CD4, APP-CD40, CD86-CTLA4, CXC, and SELPLG pathways, indicating distinctions from healthy individuals. Moreover, MIF or BAFF pathways, common in SLE and IgG4-related diseases, were not increased.

Our study underscores several statistically significant differences, yet the rarity of aHUS and the consequent limitation in case enrollment necessitates future investigations involving a broader participant base. A pertinent consideration is that aHUS’s incidence does not display a direct correlation with factors such as age or gender. However, due to funding constraints, our selection of healthy controls was restricted to four individuals and three unaffected family members. This sampling constraint may potentially introduce a selection bias, which we recognize as a limitation of our current study. We will explicitly address this point in the limitations section of our paper. Going forward, we aim to undertake more exhaustive and far-reaching studies to mitigate this issue and enhance the robustness of our findings.

The sensitivity of our scRNA-seq method, influenced by protocol specifics and data quality, generally detects thousands of genes per cell, but is limited in discerning lowly expressed genes. The complementary methods used, CellChat and Monocle 3, similarly rely on data quality and dataset characteristics. Statistical test sensitivity is tied to sample size and effect size. Despite their recognized limitations, we deemed these methods suitably sensitive for our research, bolstered by rigorous protocol adherence and multifaceted validation to ensure finding robustness and reliability.

## 4. Materials and Methods

### 4.1. Patient Recruitment

In this single-center Taiwanese study, peripheral blood samples from 13 adult aHUS patients, 3 unaffected family members, and 4 healthy subjects were analyzed using scRNA-seq. aHUS patients were classified into stable and unstable groups, receiving plasma exchange alone or combined with anti-complement therapy. Stable disease had stable TMA-related organ involvement and normal hemolysis markers.

Three individuals were unaffected family members who were directly blood-related to our aHUS patients. It was validated that they neither exhibited any clinical symptoms of aHUS nor demonstrated any anomalies in their hemolysis markers. Further supplementing our control cohort were four healthy medical professionals who willingly participated as ‘healthy controls’. Everyone underwent rigorous health evaluations, ensuring not only the absence of abnormalities in their hemolysis markers, but also confirming no personal or familial history of aHUS.

### 4.2. Single Cell RNA-Seq and Data Analysis

scRNA-seq was performed as previously described [25]. Sequencer raw data were processed using CellRanger v3.1.0 or v6.0.2 with the GRCh38-3.0.0 reference file. Cells were selected using Seurat v4.0.4 [26] in R software (www.R-project.org, accessed on 5 February 2023) based on criteria including detected genes, unique molecular identifiers (UMIs), mitochondrial gene read counts, and doublet identification using Scrublet v0.2.2 [27]. Data type conversion was performed using Scanpy v1.8.1 [28] and SeuratDisk v0.0.0.9015 tools (https://mojaveazure.github.io/seurat-disk/, accessed on 5 February 2023).

### 4.3. Single Cell RNA-Seq Data Integration and Clustering

Using the SCTransform workflow, scRNA-seq datasets were integrated, scaled, and normalized, considering regression variables such as cell cycle stage, mitochondrial reads, gene number, and UMI count. The top 3000 variable genes were selected for PCA using SelectIntegrationFeatures. A reference-based integration workflow with rPCA was applied, using four healthy control samples as reference. The top 50 PCs from PCA were used for UMAP, and the FindNeighbors function constructed a nearest neighbors graph for clustering analysis, all provided by the Seurat v4.0.4 package.

### 4.4. Cell Type Annotations

Using SingleR v1.4.1 [29], a reference-based cell type annotation tool, cell types in the dataset were classified by comparing gene expression profiles and assigning nomenclature and cell ontology terms. Reference gene expression data were obtained from five functions provided by the celldex v1.0.0 R package. MonacoImmuneData labels were selected first, followed by Macrophages M1 and M2 labels from BlueprintEncodeData. Lastly, cell types such as Macrophages (CL: 0000235), Lung Macro (CL: 0000583), INF-Macro (CL: 0000863), and Megakaryocyte (CL: 0000556) were identified using cell ontology terms.

### 4.5. Clustering Analysis

Further clustering analysis was performed using the same parameters as before, with a resolution range of 0.1 to 0.5. The Seurat function FindAllMarkers was applied to identify expression markers for each cluster in each cell type.

### 4.6. Pseudotime Estimation

Monocle 3 [30] was utilized to construct cell trajectory paths for B-cells, CD4+ T-cells, CD8+ T-cells, and monocyte lineages. This involved dimensionality reduction using PCA and UMAP, followed by Leiden clustering [31]. Trajectory paths were built by connecting nearest neighbors in the UMAP graph, with the root node determined based on early-stage cell types, such as naïve B-cells, naïve T-cells, and monocytes. Cell type abundance along pseudotime from the trajectory path was also analyzed.

### 4.7. Cell–Cell Communication Analysis

The CellChat v1.1.3 [32] R package was used to infer the probability of ligand-receptor signaling communication among all cell types in scRNA-seq datasets. Heatmaps of cell–cell interaction probability were generated for each sample using the ComplexHeatmap [33] R package and visualized with Morpheus (https://software.broadinstitute.org/morpheus, accessed on 5 February 2023) for each signaling pathway.

### 4.8. Statistical Analysis

The abundance of cell types and further clusters are presented as percentages. Continuous data were compared using Wilcoxon rank sum tests with the R package ggpubr (v0.4.0) (https://rpkgs.datanovia.com/ggpubr, accessed on 5 February 2023), and confidence intervals were calculated using the R package asht (v0.9.7) [34].

### 4.9. Ethics Approval and Consent to Participate

In accordance with ethical standards, we obtained informed consent from all participants. This study received approval from the Research Ethics Committee of China Medical University Hospital, Taiwan (CMUH111-REC2-048). Additionally, all procedures adhered to both the principles of the Declaration of Helsinki and the Good Clinical Practice Guidelines. Every participant provided their explicit informed consent before partaking in this study.

## 5. Conclusions

Our study presents robust evidence underscoring the critical role of immune cell dysregulation in the pathogenesis of atypical hemolytic uremic syndrome (aHUS). We reveal intermediate monocytes as a novel potential disease activity marker, displaying significant abundance in unstable aHUS cases. Furthermore, our investigation into immune cell subclusters exposes distinct gene expression profiles between stable and unstable aHUS, thus offering potential clinical indicators for assessing aHUS disease activity. Our pseudotime trajectory analysis uncovers a divergence point in immune cell differentiation between aHUS patients and healthy controls, highlighting a previously unobserved pathogenetic mechanism. Complementing this, our cell–cell interaction analysis discloses striking differences in signaling pathways between aHUS patients and healthy individuals, suggesting altered intercellular communication as a crucial player in aHUS. Taken together, these findings significantly enrich our understanding of the molecular mechanisms underpinning aHUS and hold the potential to catalyze the development of novel diagnostic tools and disease activity markers. We posit that this study paves the way for future investigations into the pathogenesis of aHUS and the development of innovative therapeutic strategies for this complex disease.

## Figures and Tables

**Figure 1 ijms-24-10007-f001:**
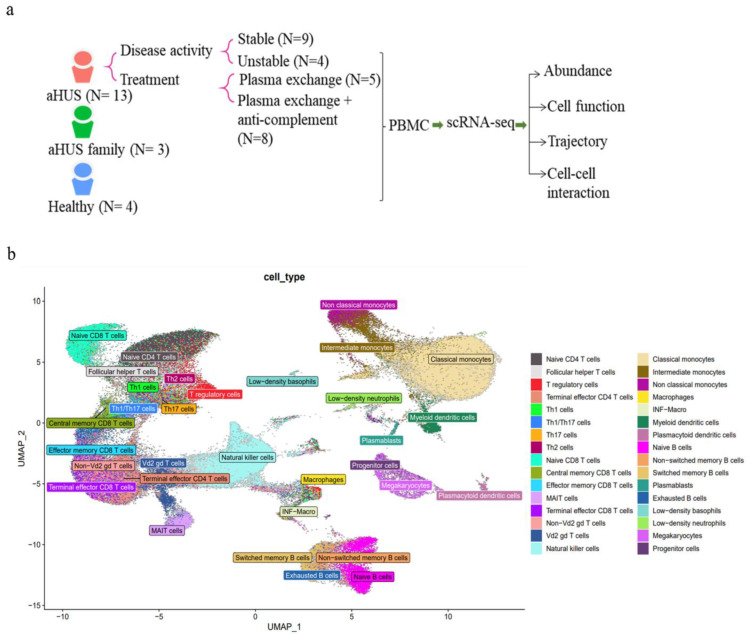
The immune cell phenotype of aHUS patients, aHUS family, and healthy controls were investigated using scRNA-seq analysis of PBMCs. The study design is presented in (**a**). UMAP coordinates showing the distribution of immune cells in PBMCs are presented in (**b**).

**Figure 2 ijms-24-10007-f002:**
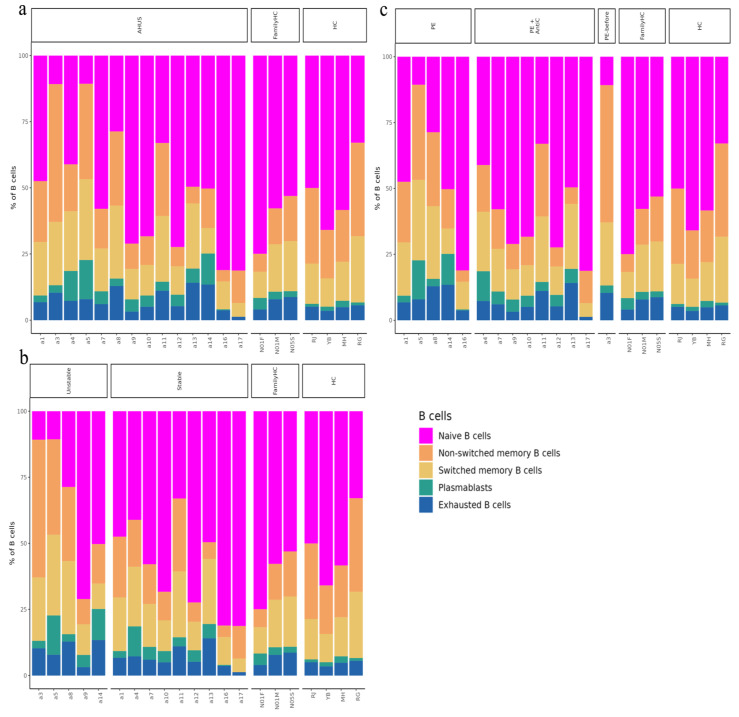
The relative abundance of subpopulations of B-cells is shown in (**a**), while the impact of aHUS disease activity on B-cell subpopulations is presented in (**b**), and the influence of aHUS treatment on B-cell subpopulations is shown in (**c**). The identifiers N01F, N01M, and N05S correspond to the individual cases of three aHUS family members included in our study. Similarly, the identifiers RJ, YB, MH, and RG denote the individual cases of four healthy controls in our study.

**Figure 3 ijms-24-10007-f003:**
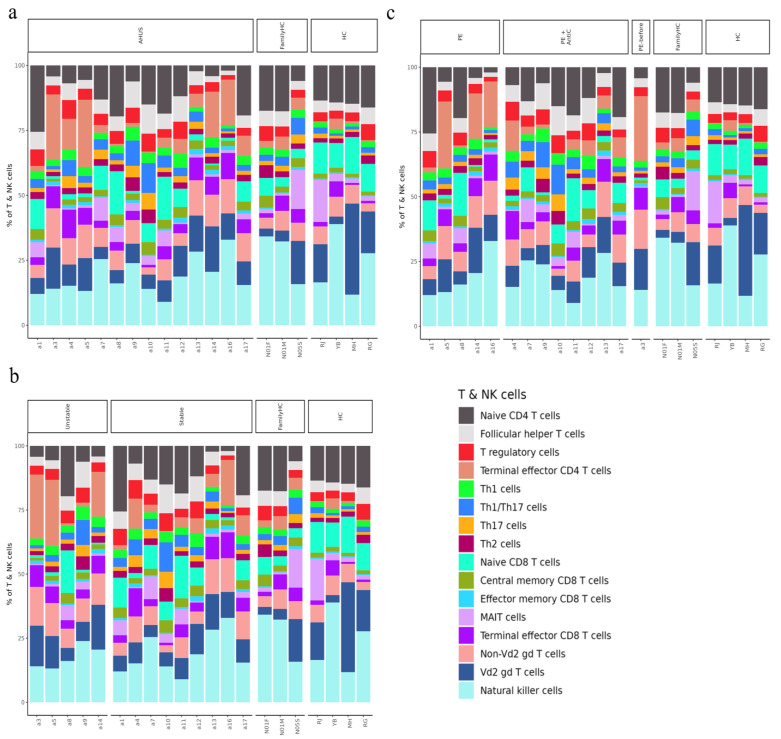
The relative abundance of subpopulations of T- and natural killer (NK) cells is shown in (**a**), the impact of aHUS disease activity on T- and NK cell subpopulations is presented in (**b**), and the influence of aHUS treatment on T- and NK cell subpopulations is shown in (**c**).

**Figure 4 ijms-24-10007-f004:**
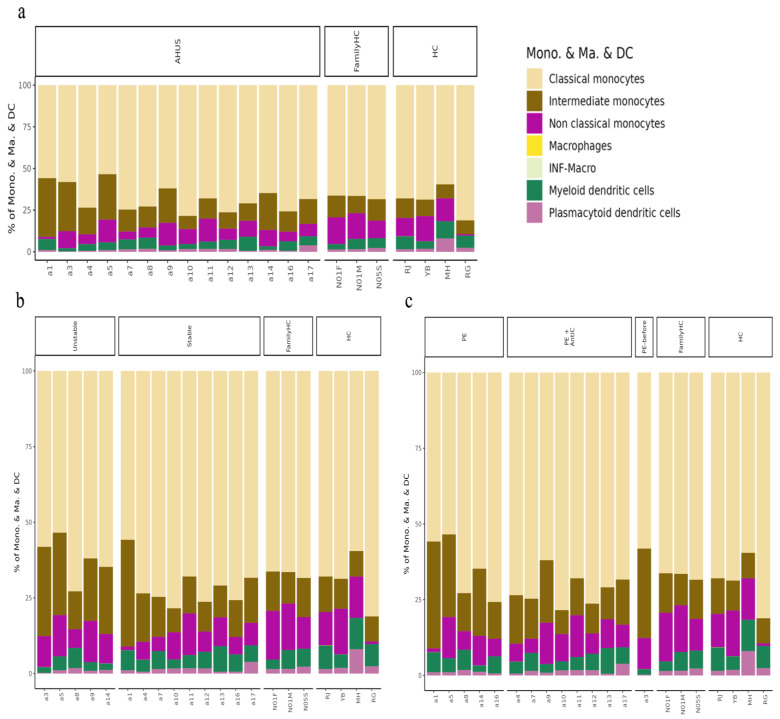
The analysis of monocyte series, including monocytes (Mono.), macrophages (Ma.), and dendritic cells (DC), is presented in (**a**), while the impact of aHUS disease activity on these cells is presented in (**b**), and the influence of aHUS treatment on these cells is shown in (**c**).

**Figure 5 ijms-24-10007-f005:**
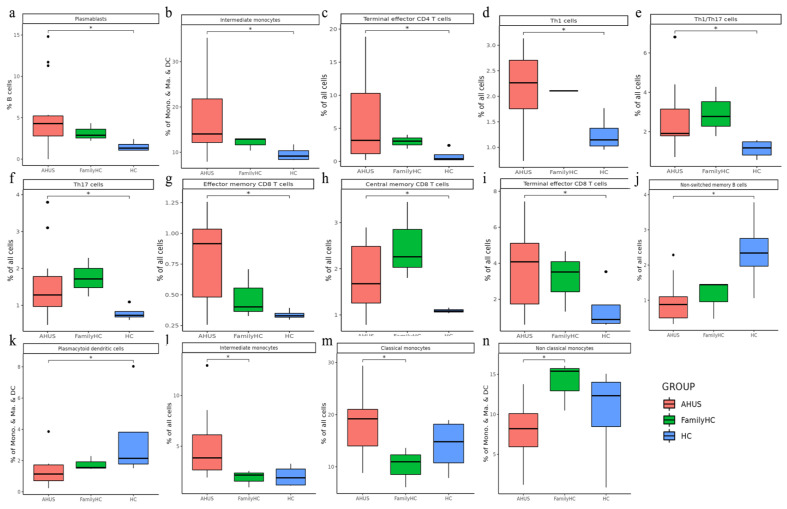
Significant abundance of cell subpopulations in PBMCs from aHUS, aHUS family, to healthy subjects (**a**–**n**) * *p* < 0.05.

**Figure 6 ijms-24-10007-f006:**
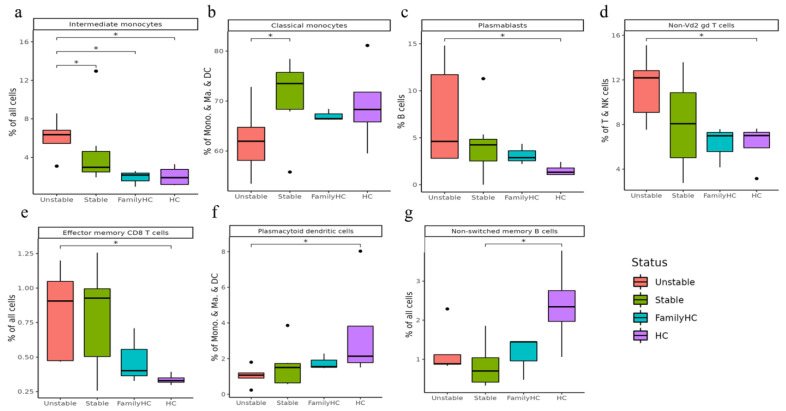
Significant abundance of cell subpopulations in PBMCs from aHUS in unstable and stable disease activity, aHUS family, to healthy subjects (**a**–**g**). * *p* < 0.05.

**Figure 7 ijms-24-10007-f007:**
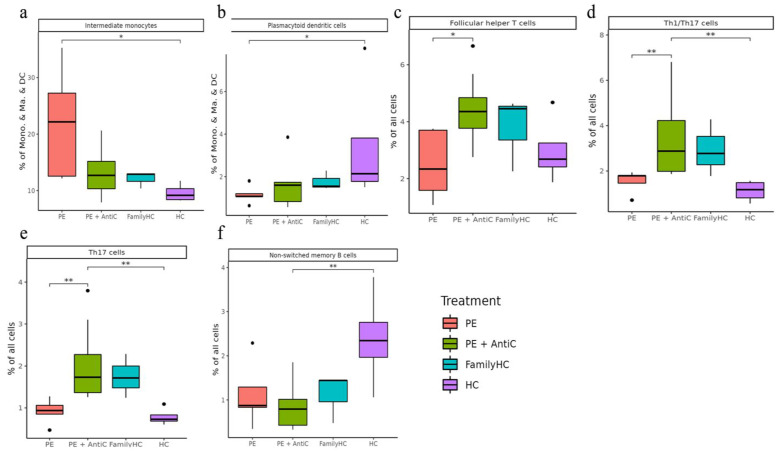
Significant abundance of cell subpopulations compared in PBMCs from aHUS treatment with plasma exchange alone, treatment combined with plasma exchange and anti-complement therapy, to aHUS family and healthy subjects (**a**–**f**) * *p* < 0.05; ** *p* < 0.01.

**Figure 8 ijms-24-10007-f008:**
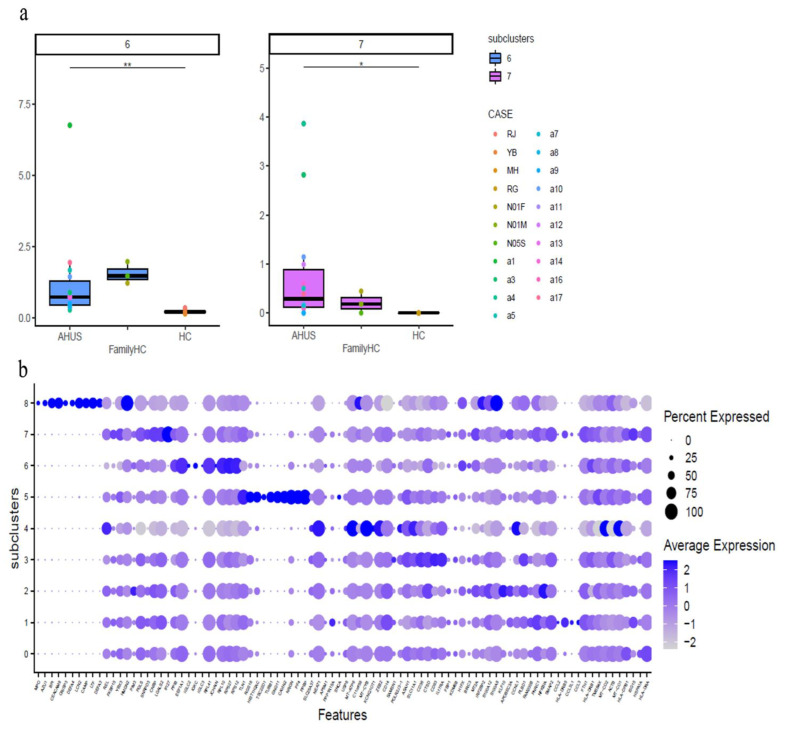
The figure presents boxplots displaying the subcluster significant abundance of classical monocytes (**a**) in PBMCs of individuals with aHUS, aHUS family, and healthy subjects. (**b**) Dot plots of the gene expression profiles of the top 10 marker genes in each subcluster are also provided. Statistically significant differences are indicated by * *p* < 0.05 and ** *p* < 0.01.

**Figure 9 ijms-24-10007-f009:**
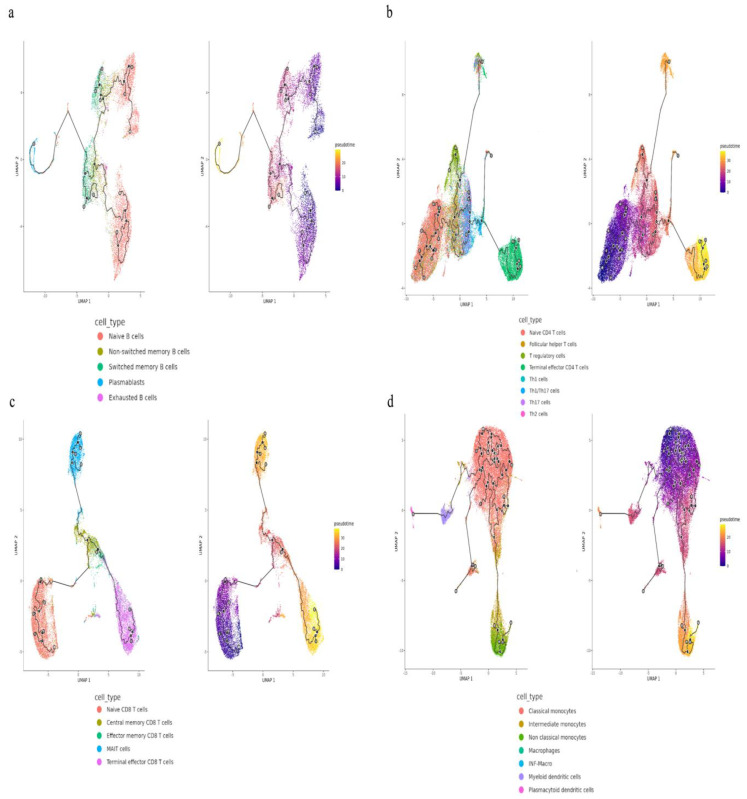
The trajectories for B-cells (**a**), T-cells (**b**,**c**), and monocytes (**d**) with different state dynamics of the immune cells.

**Figure 10 ijms-24-10007-f010:**
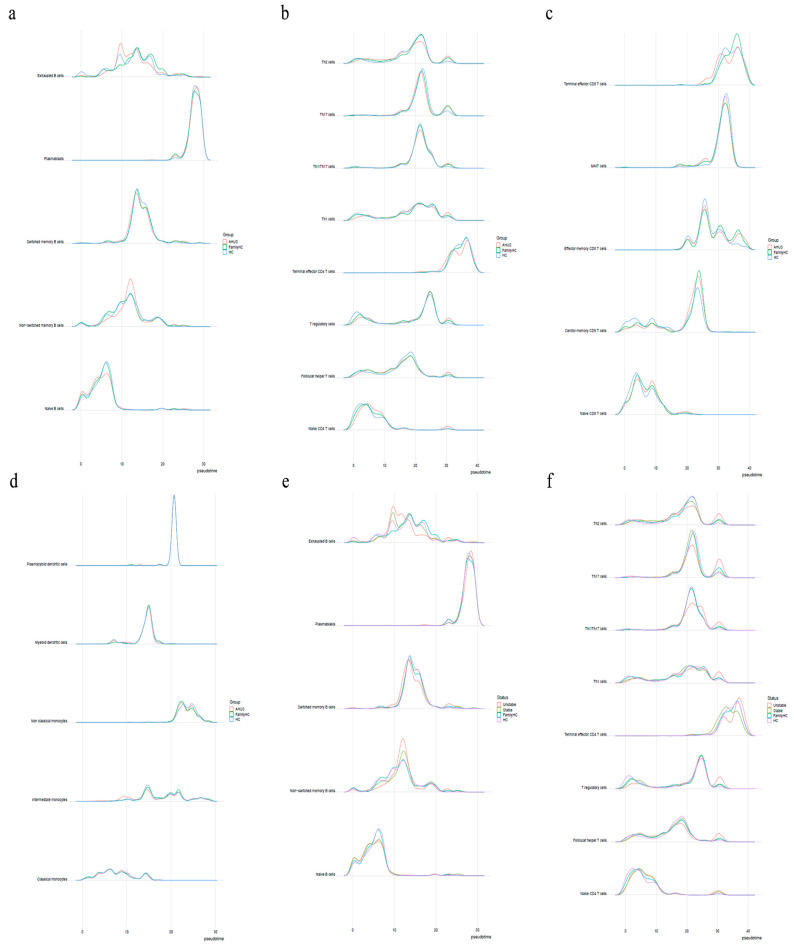
(**a**) Shows the pseudotime interval difference and abundance of B-cells, while (**b**,**c**) show the pseudotime interval difference and abundance of T-cells. (**d**) The pseudotime interval difference and abundance of monocytes. The pseudotime interval difference and abundance of B-cells (**e**) and T-cells (**f**), respectively, among the unstable aHUS group, stable aHUS group, aHUS family, and healthy subjects.

**Figure 11 ijms-24-10007-f011:**
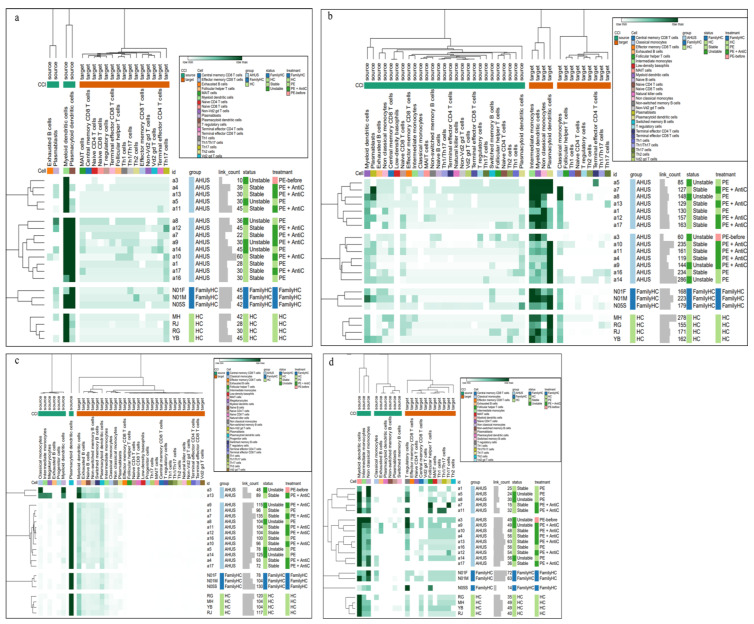
Cell–cell interaction signaling of ALCAM-CD6 (**a**), IL16-CD4 (**b**), APP-CD40 (**c**), CD86-CTLA4 (**d**) among individuals with aHUS with varying disease activity, treatment, aHUS family members, and healthy controls.

**Table 1 ijms-24-10007-t001:** Characteristics of 13 aHUS patients.

Case	Age	Gender	TMA Involvement Organ	Treatment	Disease Activity
a1	39	M	Kidney, brain, lung, heart	PE	Stable
a3/a5	81	M	Kidney, heart	a3 before 1st PE a5 after PE	Unstable
a4	66	M	Kidney, brain, heart	PE + anti C5	Stable
a7	36	F	Kidney, heart, pancreas, eye	PE + anti C5	Stable
a8	33	F	Kidney, brain, lung, heart	PE	Unstable
a9	62	F	Kidney, brain, heart	PE + anti C5	Unstable
a10	39	M	Kidney, brain, lung, heart, eye	PE + anti C5	Stable
a11	30	F	Kidney, brain, lung, heart, eye, bowel	PE + anti C5	Stable
a12	42	F	Kidney, brain, lung, heart, pancreas, liver, eye, skin	PE + anti C5	Stable
a13	53	M	Kidney, brain, heart	PE + anti C5	Stable
a14	70	M	Kidney, brain, heart	PE	Unstable
a16	62	F	Kidney, heart	PE	Stable
a17	49	F	Kidney, brain, lung, heart, pancreas, liver, eye	PE + anti C5	Stable

F, female; M, male; TMA, thrombotic microangiopathy; PE, plasma exchange; anti-C5, anti-complement therapy.

**Table 2 ijms-24-10007-t002:** Significant difference in cell subclusters in aHUS and healthy controls.

**Significantly Increased Immune Cell Subclusters in aHUS Patients Compared to Healthy Controls with Correlated Gene Expression Increasing**			
Cell Subclusters		*p* value	Higher expression levels of gene
Classical monocyte subclusters 6		*p* < 0.01	RPS27
Classical monocyte subclusters 7		*p* < 0.05	IFI27
Central memory CD8 T-cells subcluster 3		*p* < 0.05	CXCR4
Non-Vd2 gd T-cells subcluster 4		*p* < 0.05	SYNE2
Th1 cells subcluster 3		*p* < 0.05	MT-CYB
Th17 cells subcluster 4		*p* < 0.05	MT-ATP6
**Significantly Increased Immune Cell Subclusters in Healthy Controls Compared to aHUS Patients with Correlated Gene Expression Increasing**			
Cell Subclusters		*p* value	Higher expression levels of gene
Central memory CD8 T-cells subcluster 1		*p* < 0.05	EIF3E
Th1 cells subcluster 0		*p* < 0.05	RPS27
Non classical monocytes subcluster 5		*p* < 0.01	LYPD2
Terminal effector CD4 T-cells subcluster 3		*p* < 0.01	KLRD1
Th17 cells subcluster 3		*p* < 0.05	ACTG1, CD52 and LGALS1

**Table 3 ijms-24-10007-t003:** Significant difference in cell subclusters in aHUS disease activity.

Significantly Increased Immune Cell Subclusters in Unstable aHUS Patients Compared to Stable aHUS Patients with Correlated Gene Expression Increasing			
Cell Subclusters		*p* value	Higher expression levels of gene
Classical monocyte subclusters 4		*p* < 0.05	NEAT1, MT-ATP6 and MT-CYB
Central memory CD8-T cells subcluster 2		*p* < 0.05	VIM
Non-Vd2 gd T-cells subcluster 1		*p* < 0.05	ACTG1
Terminal effector CD8—cells subcluster 3		*p* < 0.05	RPL13
Terminal effector CD8 T-cells subcluster 5		*p* < 0.01	KLRB1
**Significantly Increased Immune Cell Subclusters in stable aHUS Patients Compared to unstable aHUS Patients with Correlated Gene Expression Increasing**			
Cell Subclusters		*p* value	Higher expression levels of gene
Central memory CD8 T-cells subcluster 1		*p* < 0.05	RPL23
Non-Vd2 gd T-cell subcluster 0		*p* < 0.05	GZMH
Th1 cells subcluster 0		*p* < 0.05	RPS27, RPS4X

## Data Availability

Data will be made available upon request to the corresponding author.

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
