# Peer review of "Dysregulation of Immune Cell Subpopulations in Atypical Hemolytic Uremic Syndrome"

_ijms, 2023, doi:10.3390/ijms241210007_

Round 1

Reviewer 1 Report

1.      The title needs to be changed to summarize the whole study and look the work novel.

2.      Abstract section a number of abbreviations though used but not defined. The results are also not mentioned in the numerical form.

3.      Keywords section: arrange them alphabetically and omit those which are part of manuscript title.

4.      Introduction section: it is very short and do not provide an insight into the subject matter to the readers. The introduction should be at least three paragraphs. In the first paragraph explain the problem under study. The second about its literature whereas the third should explain the research gap and aims and objectives of the study.

5.      Table 1 result section in first column what “a” signifies.

6.      Figure 1 seems somewhat blurred please increase its resolution.

7.      Figure 3 is also not clear.

8.      Discussion: the results should be properly compared with those reported in literature. Also, what was the sensitivity of the method applied.

9.      Ethical statement is absent. The study involves human therefore, consents and guidelines followed should be clearly mentioned.

10.    Conclusions needs to be rephrased.

Minor changes required

Author Response

Dear Reviewer,

Thank you for your time and insightful comments on our manuscript. Your feedback is highly appreciated and has helped us to significantly improve our work. We have addressed each of your comments as follows:

Point 1: The title needs to be changed to summarize the whole study and look the work novel.

Response 1: Regarding reviewer’s comment about the title, we have revised it to "Dysregulation of Immune Cell Subpopulations in Atypical Hemolytic Uremic Syndrome". We believe this title better summarizes our study and highlights its novelty.

Point 2: Abstract section a number of abbreviations though used but not defined. The results are also not mentioned in the numerical form.

Response 2: We acknowledge the issues you have pointed out in the abstract section, particularly with respect to the undefined abbreviations and the lack of numerical results. We understand the necessity for clarity and have revised the abstract accordingly.

We have now defined all abbreviations at their first mention in the abstract and ensured the inclusion of numerical results to better present our findings.

Point 3: Keywords section: arrange them alphabetically and omit those which are part of manuscript title

Response 3: We have revised the keywords section and removed any keywords that are part of the manuscript title.

We believe these changes will make the manuscript easier to locate in searches and are a better reflection of the content of our paper.

Point 4: Introduction section: it is very short and do not provide an insight into the subject matter to the readers. The introduction should be at least three paragraphs. In the first paragraph explain the problem under study. The second about its literature whereas the third should explain the research gap and aims and objectives of the study.

Response 4: We appreciate the reviewer's insights and have fully updated our Introduction section in accordance with your suggestions.

Point 5: Table 1 result section in first column what “a” signifies.

Response 5: The "a" in the first column of the results section in Table 1 serves as the identifier for each individual patient with the rare disease aHUS that was included in our study. We have used the lowercase "a" from aHUS as the identifier for clarity and consistency."

Point 6: Figure 1 seems somewhat blurred please increase its resolution.

Response 6: We have now enhanced the resolution of all images for clearer visualization. Given that this optimization has increased the total number of images, we have consequently moved some of these improved figures to the supplementary material.

Point 7: Figure 3 is also not clear.

Response 7: We have subsequently increased the resolution of all our images for better clarity. However, this enhancement resulted in an increased quantity of images, leading us to relocate some of the updated figures to the supplementary materials.

Point 8: Discussion: the results should be properly compared with those reported in literature. Also, what was the sensitivity of the method applied.

Response 8: We greatly appreciate your thoughtful comments and suggestions concerning our manuscript. In response to your observations, we have revised our discussion section to include a comprehensive comparison of our results with those reported in existing literature. This revision offers an insightful context for our findings, and we believe it enriches the overall discussion.

Regarding the sensitivity of the applied methods, we would like to clarify the following:

The sensitivity of single-cell RNA sequencing (scRNA-seq) largely depends on various protocol specifics, such as cell capture efficiency, reverse transcription efficiency, library construction, and sequencing depth. As you pointed out, scRNA-seq generally detects the expression of thousands of genes per cell but has limitations in detecting lowly expressed genes. This is a widely acknowledged limitation of scRNA-seq. In our study, the precise sensitivity of the scRNA-seq protocol, including details about the detection of specific transcripts or lowly expressed genes, has been previously described in the referenced paper [25], and we have adhered to their protocol to maintain the sensitivity.

For the CellChat method, the sensitivity is dependent on the complexity and quality of the input data. We have taken extensive measures to ensure optimal data quality for our analysis.

For Monocle 3, the sensitivity in determining pseudotime is contingent on the quality of the dataset and the structure of the differentiation trajectory.

In terms of the statistical tests we used, the sensitivity can be associated with the sample size and effect size under investigation.

In summary, while acknowledging the inherent limitations of these methods, we are confident that they provided appropriate sensitivity for our study objectives. We have attempted to corroborate our results using multiple analytical methods to ensure robustness and reliability.

We have now included these points in the revised discussion to provide clarity and enhance understanding of the sensitivity of the methods applied in our study.

Once again, we thank you for your insightful suggestions, which have undeniably improved the quality of our manuscript.

Point 9: Ethical statement is absent. The study involves human therefore, consents and guidelines followed should be clearly mentioned.

Response 9: We have now included the 'Ethics approval and consent to participate' section in 4.9 of our manuscript.

Point 10: Conclusions needs to be rephrased.

Response 10: We appreciate the reviewer's insights and have fully rephrase our Conclusions section.

Point 11: Moderate editing of English language is recommended.

Response 11: We sincerely appreciate the reviewer's suggestion. Please note that our revised manuscript has been thoroughly reviewed and polished by the Language Editing Services recommended by MDPI prior to this resubmission.

Reviewer 2 Report

Major comments:

1. Some key information about the aHUS needs to be provided in the introduction, i.e. what are some of the known genetic variations for the aHUS, what are the clinical characteristics of stable and unstable aHUS, what is the basis of anti-complement therapy in aHUS, etc. This may help readers from outside the aHUS field understand the results better.

2. The criteria for the control group donors selection needs to be further elucidated. In this study, the sample size of unaffected family members and healthy donors are relatively small comparing to aHUS patients. Are the selected subjects unbiased and sufficient to represent the two unaffected populations?

3. A major problem for all figures in this manuscript is the text sizes are too small and the resolutions are not high enough, making it extremely difficult to read the figures. One general suggestion is to reduce some less-relevant figures and move them into supplementary materials. 

4. Stable and unstable aHUS were analyzed together as one group in some results, while were analyzed as separate groups in other parts of results. Could the authors justify the inconsistency?   

Minor comments:

1. In Table 1 legend, please indicate what "TMA" stands for.

2. In Figure 2d, data for "family HC" seems to be without a variation, is this the fact or an error?

Minor editing of English language is recommended.

Author Response

Dear Reviewer,

Thank you for your time and insightful comments on our manuscript. Your feedback is highly appreciated and has helped us to significantly improve our work. We have addressed each of your comments as follows:

Point 1: Some key information about the aHUS needs to be provided in the introduction, i.e. what are some of the known genetic variations for the aHUS, what are the clinical characteristics of stable and unstable aHUS, what is the basis of anti-complement therapy in aHUS, etc. This may help readers from outside the aHUS field understand the results better.

Response 1: We appreciate the reviewer's insights and have fully updated our Introduction section in accordance with your suggestions.

Point 2: The criteria for the control group donors selection needs to be further elucidated. In this study, the sample size of unaffected family members and healthy donors are relatively small comparing to aHUS patients. Are the selected subjects unbiased and sufficient to represent the two unaffected populations?

Response 2: We appreciate the reviewer's query. For our study, the control group was selected as follows: three unaffected family members served as part of the control group. These individuals were direct blood relatives of our aHUS patients but were confirmed to not show any clinical manifestations of aHUS, nor did they exhibit any abnormalities in their hemolysis markers. Additionally, four healthy medical workers voluntarily participated as 'healthy controls'. These individuals underwent health checks and were confirmed to not only lack any abnormalities in their hemolysis markers but also have no personal or familial history of aHUS. Given that the occurrence of aHUS is not directly associated with age or gender, our selection of healthy controls was limited to four individuals and three unaffected family members due to budget constraints. Indeed, this could potentially introduce bias in the selection of subjects. We plan to acknowledge this as a limitation in our study. Additionally, we look forward to conducting more comprehensive and extensive studies in the future to address this issue.

Point 3: A major problem for all figures in this manuscript is the text sizes are too small and the resolutions are not high enough, making it extremely difficult to read the figures. One general suggestion is to reduce some less-relevant figures and move them into supplementary materials.

Response 3: We have now enhanced the resolution of all images for clearer visualization. Given that this optimization has increased the total number of images, we have consequently moved some of these improved figures to the supplementary material.

Point 4: Stable and unstable aHUS were analyzed together as one group in some results, while were analyzed as separate groups in other parts of results. Could the authors justify the inconsistency?

Response 4: We appreciate the reviewer's query. Our analysis initially involves a comparison among aHUS patients, unaffected family members of aHUS patients, and healthy controls. Subsequently, to probe the variations in aHUS disease activity, we segregate aHUS patients into stable and unstable subgroups. Furthermore, to verify whether the observed outcomes are a result of different therapeutic approaches, we further stratify our 13 aHUS cases into two treatment modalities: those undergoing only plasma exchange and those receiving plasma exchange supplemented with anti-complement therapy.

Point 5: In Table 1 legend, please indicate what "TMA" stands for.

Response 5: We are grateful for the reviewer's astute observation. The acronym "TMA" represents Thrombotic Microangiopathy. To clarify this for all readers, we have now added this explanation to the figure legend. We believe this amendment will enhance the comprehension of our graphical data.

Point 6: In Figure 2d, data for "family HC" seems to be without a variation, is this the fact or an erro

Response 6: We value the reviewer's keen observation. Indeed, this observation aligns with the facts presented in our study.

Point 7: Minor editing of English language is recommended.

Response 7: We sincerely appreciate the reviewer's suggestion. Please note that our revised manuscript has been thoroughly reviewed and polished by the Language Editing Services recommended by MDPI prior to this resubmission.

Round 2

Reviewer 1 Report

ok

ok

Reviewer 2 Report

The introduction, data presentation, conclusion and English-language are well-improved from the first draft. The current version is acceptable for publication.